# Specific Learning Disabilities and Emotional-Behavioral Difficulties: Phenotypes and Role of the Cognitive Profile

**DOI:** 10.3390/jcm12051882

**Published:** 2023-02-27

**Authors:** Paola Cristofani, Maria Chiara Di Lieto, Claudia Casalini, Chiara Pecini, Matteo Baroncini, Ottavia Pessina, Filippo Gasperini, Maria Bianca Dasso Lang, Mariaelisa Bartoli, Anna Maria Chilosi, Annarita Milone

**Affiliations:** 1Department of Developmental Neuroscience, IRCCS Fondazione Stella Maris, 56128 Pisa, Italy; 2Department of Education, Languages, Intercultures, Literatures and Psychology, University of Florence, 50121 Firenze, Italy; 3Department of Clinical and Experimental Medicine, University of Pisa, 56126 Pisa, Italy

**Keywords:** specific learning disabilities, emotional-behavioral manifestation, child behavior checklist-CBCL, learning impairment, working memory, cognitive profile

## Abstract

Specific Learning Disabilities (SLD) are often associated with emotional-behavioral problems. Many studies highlighted a greater psychopathological risk in SLD, describing both internalizing and externalizing problems. The aims of this study were to investigate the emotional-behavioral phenotype through the Child Behavior Checklist (CBCL), and evaluate the mediating role of background and cognitive characteristics on the relationship between CBCL profile and learning impairment in children and adolescents with SLD. One hundred and twenty-one SLD subjects (7–18 years) were recruited. Cognitive and academic skills were assessed, and parents completed the questionnaire CBCL 6–18. The results showed that about half of the subjects manifested emotional-behavioral problems with a prevalence of internalizing symptoms, such as anxiety and depression, over externalizing ones. Older children showed greater internalizing problems than younger ones. Males have greater externalizing problems compared to females. A mediation model analysis revealed that learning impairment is directly predicted by age and familiarity for neurodevelopmental disorders and indirectly via the mediation of the WISC-IV/WAIS-IV Working Memory Index (WMI) by the CBCL Rule-Breaking Behavior scale. This study stresses the need to combine the learning and neuropsychological assessment with a psychopathological evaluation of children and adolescents with SLD and provides new interpretative insights on the complex interaction between cognitive, learning, and emotional-behavioral phenotypes.

## 1. Introduction

Specific Learning Disabilities (SLD) are neurodevelopmental disorders characterized by difficulties in reading, writing, and/or mathematics skills in parallel with average intelligence and adequate educational and sociocultural opportunities, which occur in the absence of sensory, neurologic, or other psychiatric disorders [1]. On average, 5–15% of students are diagnosed with SLD, such as Dyslexia, Dyscalculia, Dysgraphia, and Dysorthography [1]. SLD persist despite normal schooling and can interfere with school education and daily life by having an impact on academic achievement, relational experiences, and quality of life. Furthermore, SLD can be characterized by the coexistence of emotional and behavioral problems [2,3,4,5,6,7]. In fact, a relevant feature of SLD is comorbidity [8,9]: It frequently happens to encounter the coexistence not only of several types of SLD (“homotypic comorbidity”), but also of other neurodevelopmental disorders or clinical conditions (“heterotypical comorbidity”) in the same subject [10].

The comorbidity between SLD and emotional-behavioral disorders is high: About 30% of children with SLD have emotional and behavioral problems [11], whose phenomenology can be very heterogeneous and is often described in terms of both internalizing and externalizing disorders [12]. From an empirical point of view of child behavior classification, anxiety, depression, social withdrawal, and somatic complaints are conceptualized as internalizing problems, while disinhibited or externally-focused behavioral symptoms, including aggression, conduct problems, rule breaking behavior, oppositionality, hyperactivity, are conceptualized as externalizing problems [13]. For what concerns internalizing symptoms, children with SLD show a higher rate of separation anxiety and generalized and social anxiety compared to children who do not have SLD problems [6]. Moreover, dyslexic children report higher levels of somatic symptoms [3] and depressive symptomatology [14]. Regarding externalizing symptoms, both oppositional defiant [15] and conduct disorders [16] have been reported in SLD. However, the highest comorbidity is with Attention Deficit Hyperactivity Disorder (ADHD) [17,18], occurring between 25% and 45% of SLD cases [19,20].

The interaction between emotional-behavioral functioning and learning difficulties is very complex and its explanation calls into question several hypotheses.

The association between SLD and emotional-behavioral disorders can be traced back to a causal role of the former on the latter (e.g., school failure causes emotional distress in the student) but also of the latter on the former (e.g., a subject with emotional-behavioral problems does not work hard in learning tasks, caused by different factors, such as reduced motivation, low tolerance to frustration or resistance to stress, impulsiveness). Therefore, SLD can be hypothesized as a risk factor for high levels of internalizing and externalizing symptoms, which might be considered a consequence of SLD themselves, deriving from the impact of the disorder on the mental health of SLD subjects [21]. Moreover, emotional-behavioral disorders can be hypothesized as a risk factor for SLD too.

However, the concept of comorbidity contemplates the idea that different clinical conditions arise together without any causal link between them and may be the expression of a more general malfunction that shares a common neuropsychological substrate. In this regard, the hypothesis of the “Multiple Deficit Model” [21] suggests that disorders are linked together by multiple factors, some specific to a given disorder while others are in common, and they can derive from the interaction between several risk factors [8]. Indeed, the interaction between SLD and emotional-behavioral problems seems mediated by a series of biological, cognitive, and environmental factors that document the role of many distinct and shared aspects. Willcutt and Pennington [3] found a close relationship between reading disorders and externalizing disorders in males, while females show more internalizing symptoms and, specifically, they report more depressive and somatic symptoms than males. The association between SLD and emotional difficulties seems to be mediated also by age: For example, Giovagnoli and colleagues [22] found higher levels of internalizing symptoms in schoolers with developmental dyslexia, specifically, school anxiety in those children who attended secondary school compared to the primary ones. The age of diagnosis also influences the emotional-affective aspect. An early diagnosis allows the activation of more effective coping strategies for the disorder with positive effects on self-esteem and sense of self [23]. The absence of an adequate diagnosis represents a negative aspect as the person reports a sense of inability [24]. Evidence in the literature shows that children identified later as dyslexics report greater feelings of low self-esteem and fear of judgment [25]. At the same time, self-esteem seems to be higher in the presence of an early diagnosis of dyslexia [26].

Furthermore, there are still relatively few studies investigating the relationship between the cognitive and neuropsychological profile and the presence of internalizing and externalizing symptoms in SLD. Mugnaini and collaborators [27] report that a borderline cognitive level in subjects with dyslexia represents a risk factor for the presence of internalizing problems. Some further studies examined the association between cognitive profile and emotional difficulties in poor reader children without a clear diagnosis of SLD. Nachshon and Horowitz-Kraus [28] demonstrated the relationship between low reading scores, emotional abilities, and executive functions (such as speed of processing, inhibition, visual attention, and switching abilities). Another cognitive factor mediating the interaction between SLD and emotional-behavioral problems is the presence of language impairment (assessed in 20% of preschool-aged children with reading disorders) [29]. It is well-known that language disorders, besides being a risk factor for the development of literacy difficulties [30,31], seem to be a precursor for the development of emotional and behavioral problems [32,33,34,35].

Moreover, some studies report that even the extent of learning difficulties could be associated with different manifestations of emotional-behavioral difficulties. Prior and collaborators [12] found that impairment of a single learning domain is usually associated with internalizing disorders, while the presence of impairment of multiple learning domains is more frequently associated with externalizing disorders.

In summary, the interaction between emotional-behavioral functioning and learning difficulties in SLD could be complex and affected by multiple risk factors. Thus, further investigations are needed in order to shed light on the characteristics of the emotional-behavioral disorders in subjects with SLD and on the mediator role of several other factors, such as cognitive and background factors. Specifically, it would be appropriate to better understand how these factors influence the relationship between emotional and behavioral clinical manifestations and learning disorders. Such knowledge will allow better understanding of the symptom manifestation and severity of the clinical picture of children with SLD to better support the clinical management of these disorders [10,36].

Our study is part of this line of research with a dual purpose: (a) To describe the psychopathological profile of a clinical sample of subjects with SLD, (b) to investigate the mediator role of sex, age, and cognitive indices on the relationship between emotional and behavioral manifestations and learning impairment in children and adolescents with SLD. According to the literature, we hypothesize individuals with SLD show more internalizing than externalizing problems. Furthermore, regarding individual background factors, based on previous studies, we expect that males have more externalizing symptoms than females and that older age is associated with more internalizing symptoms.

## 2. Materials and Methods

### 2.1. Participants

The study included 121 subjects (73 males and 48 females, age range: 7–18 years, mean age: 11.9 ± 3.03 years) with SLD diagnosis, in accordance with the international diagnostic classification criteria [1,37] and the Italian Clinical Recommendations on SLD [38], recruited from the Specific Learning Disorder Reference Centre of the Stella Maris Foundation Hospital (Pisa, Italy). The recruitment of participants was conducted within the standard clinical care of the service, and no additional compensation was required.

Criteria for participants’ inclusion [1,38] were: (i) Age between 7 and 18 years old, (ii) second grade of primary school onward, (iii) absence of severe hearing and/or visual deficits, epilepsy, cerebral palsy, encephalitis or other major neurological abnormalities or neurodevelopmental disorders such as autism spectrum disorder, intellectual disabilities, and genetic syndromes.

All the selected subjects had the following results at the assessment carried out with standardized instruments:Both Perceptual Reasoning Index (PRI) and Verbal Comprehension Index (VCI) at the WISC-IV or WAIS-IV intelligence scales [39,40,41,42] within 1.5 standard deviation (SD) from the mean;A deficient score (z score ≤−2 or percentage rank ≤5°) in at least two reading aloud tests and/or dictation test and/or mathematics test.

Based on DSM-5 criteria, SLD can affect one or more of the following academic domains: Reading decoding, as in the case of Developmental Dyslexia, written expression or mathematics, as in the case of Developmental Dysorthography or Dyscalculia, respectively [1]. When mathematics difficulties were associated with both reading and spelling disorders, usually the most frequently encountered in clinical practice, a diagnosis of “Mixed Disorder of Scholastic Skills” was given [43].

Age of first diagnosis ranged from about 7 to 18 years and correlated with chronological age (r(120) = 0.72, *p* < 0.001), indicating that the older subjects examined in the study were being diagnosed for the first time, receiving a later diagnosis. Forty-nine percent of the participants were attending primary school (third, fourth, and fifth grade), 27% the middle school, and 24% high school. Twenty-two percent of children (26/121) had a previous psychopathological diagnosis: 10 children (8%) presented ADHD, 14 children had anxiety or depressive problems, and for 2 children the emotional-behavioral problems were not specified. All subjects were native Italian speakers of Caucasian ethnicity. This research project was approved by the Paediatric Ethics Committee of Tuscany. All parents gave written consent for their son or daughter’s participation and for publication of the results.

### 2.2. Assessment Procedure

Parents completed the Child Behavior Checklist (CBCL) [44,45], an anamnestic interview and a questionnaire on early development [46].

Each child was individually assessed by intelligence and learning tests in a quiet room with adequate lighting and a comfortable seat.

#### 2.2.1. Child Behavior Checklist (CBCL)

The presence and type of emotional and behavioral problems were explored through the Italian version of the CBCL [47,48,49]. It is a standardized scale of 118 items to be completed by the parents of children and adolescents aged between 6 and 18 years. Each item was scored on a 3-step response scale (completely agree, I do not know, completely disagree) and the items were aggregated in 8 main syndrome subscales (Anxious/Depressed, Withdrawn/Depressed, Somatic Complaints, Social Problems, Thought Problems, Attention Problems, Rule-Breaking Behaviors, Aggressive Behaviors) and six DSM-oriented sub-scales (Affective Disorder, Anxiety Disorder, Somatic Problems, Attention Deficit/Hyperactivity Problems, Oppositional Defiant Problems, Conduct Problems).

All raw scores for each CBCL scale and subscale were expressed as T scores (mean: 50; SD: 10) with respect to the age and sex norms using Achenbach’s normative data [44]. T-scores below 64 were considered normal, those between 65 and 69 in the borderline range, and those higher than 70 were in the clinical range. Three composite syndrome scales were also computable: Internalizing scale (sum of Anxious/Depressed, Withdrawn/Depressed, and Somatic Complaints), Externalizing scale (sum of Aggressive Behavior and Rule-Breaking Behavior), and Total Problem scale (sum of all syndrome subscales). For all composite scales, T-score ranges for all composite scales were as follows: Below 59 was considered normal, between 60 and 63 was borderline, and higher than 64 was clinical. Moreover, the Child Behavior Checklist Dysregulation Profile (CBCL-DP) was assessed using the sum of T-scores of the following CBCL subscales: Anxious/Depressed, Attention Problems, and Aggressive Behavior syndrome [45,50]. The combination of behavioral, affective, and cognitive dysregulation measured by the CBCL-DP was associated with an increased likelihood of psychopathology and psychosocial impairment in childhood [51]. The clinical CBCL-DP score range was ≥210.

#### 2.2.2. Intelligence

The Wechsler Intelligence Scale for Children–Fourth Edition (WISC-IV) [39] and the Wechsler Intelligence Scale for Adults–Fourth Edition (WAIS-IV) [40] were administered to evaluate the subjects’ cognitive profiles. The scales provided a full-scale IQ (FIQ) and four composite scores relating to specific cognitive abilities: (a) The Verbal Comprehension Index (VCI) measured the use and understanding of language and evaluated abstraction skills, generalization, practical reasoning, and long-term memory recovery, (b) the Perceptual Reasoning Index (PRI) assessed nonverbal reasoning and problem-solving and the ability to collect, organize, and interpret visual data to solve complex cognitive problems, (c) the Working Memory Index (WMI) measured subject’s ability to recall and manipulate auditory information in short-term memory, (d) the Processing Speed Index (PSI) included timed activities that required the analysis of visual material, visual perception, and visual scanning and hand-eye coordination. Normative data and psychometric properties for the Italian population are available [41,42].

#### 2.2.3. Academic Abilities

The academic abilities were assessed by reading decoding and comprehension, writing, and arithmetic tests described below.

Reading decoding was individually evaluated through tests of reading aloud a passage with a 4-min time limit [52,53,54], and words and non-words [54,55]. In each test, both decoding speed (i.e., in syllables for second), later expressed as z scores with reference to normative values, and accuracy (i.e., number of errors), converted to percentile rank, were computed and converted on a 3-point scale (Deficient, Borderline and Normal).

Reading comprehension was assessed through a text comprehension test, in which the subject had to answer 10 multiple-choice questions, each with four alternatives, after reading a story silently with no time limit [52,53,54]. The number of correct responses was transformed into ordinal scores on a 4-point scale (Deficient, Borderline, Within Average, and Optimal) based on normative values.

As for writing skills, they were investigated through the writing under dictation of a brief passage [54,56] and of twelve sentences containing words with unpredictable spelling only on the basis of Italian phoneme-to-grapheme correspondence rules [55]. For each test, the total number of spelling errors was computed and converted to a percentile rank according to normative value. Each percentile rank was transformed into an ordinal score on a 3-point scale (Deficient, Borderline, and Normal).

Finally, also arithmetic abilities were evaluated through standardized tests [54,57]. For each test, we converted the scores to a percentile rank according to normative value and transformed them into ordinal scores on a 3-point scale (Deficient, Borderline, and Normal).

### 2.3. Statistical Analysis

Statistical Package for Social Sciences, version 22.0 (IBM SPSS Statistics, IBM Corporation, Armonk, NY, USA) was used for statistical analyses. For missing data, case-wise deletion was applied.

Given the use of ordinal scales for the academic tests, the learning profile, and the effect of familiarity for neurodevelopmental disorder, SLD type, and sex were investigated through the non-parametric Mann–Whitney and Wilcoxon tests. Accordingly, the relation between performances at the academic tests and chronological age was investigated by non-parametric Spearman correlation.

Because of only moderate deviations from normality for the WISC-IV indices and the CBCL scales (value for both skewness and kurtosis were between −1 and +1 for each of these variables, with only 1 exception out of 19) [58] and the “robustness” of the ANOVA to non-normality of data distribution [59], parametric analyses were performed. Sphericity of data was evaluated through Muchly’s test, and in those cases where it was not confirmed, Epsilon correction on degrees of freedom of both numerator and denominator was adjusted in order to maintain the F statistic valid.

The presence of discrepancies within the cognitive and psychopathological profile, respectively, was investigated by descriptive statistics and repeated Measures ANOVAs, with Bonferroni post-hoc corrections for multiple comparisons on the Wechsler indices and the CBCL scales and subscales.

The effect of sex, familiarity for neurodevelopmental disorder, and SLD type on the scores at the Wechsler and CBCL scales and subscales was assessed through a series of between-groups ANOVAS. In order to account for potential inflation of the Type I error, Bonferroni adjusted alpha levels were used for each set of comparisons. For each of the performed ANOVAs, the effect size for the independent variable was measured by calculating Partial Eta Squared statistics.

Parametric correlation (Pearson) and regression analyses were used to investigate whether demographic and cognitive scores correlated and predicted inter-subject variability at the CBCL and learning profiles.

Based on the results from the regression analysis, we tested a mediation model (SPSS; Process v. 4.0, Model 4) on the relationship between the main background, cognitive, learning, and emotional-behavioral variables.

## 3. Results

### 3.1. Academic and Cognitive Profile

Descriptive statistics of the score range obtained at the academic test are reported in Table 1. The SLD type included Developmental Dyslexia and/or Dysorthography (51%, 62/121), Developmental Dyscalculia (6%, 8/121), and Mixed SLD (42%, 51/121). The group with Mixed SLD showed a higher (z = −5.65, *p* < 0.001) median impairment than the groups with isolated SLD (Dyslexia, Dysorthography, or Dyscalculia).

Cognitive characteristics of the SLD sample are reported in Table 2. Missing data concern the cognitive assessment conducted in another clinical center. In particular, all cognitive scores for three subjects, the WMI and PSI scales for three children, and the CVI scale for one child were not available.

A significant difference across WISC-IV indices was found (F(3, 339) = 49.52, *p* < 0.001), with a large effect-size (Partial Eta Squared = 0.30). Post-hoc analyses revealed the absence of significant differences between VCI and PRI, while significant better performances were found for both VCI and PRI with respect to both WMI and PSI (*p* < 0.05). A significant difference between WMI and PSI was also found in favor of the latter (*p* < 0.05) (VCI = PRI > PSI > WMI). In the appendix (Appendix A), t statistics for Bonferroni post-hoc corrections are reported.

Accordingly, borderline (<85) or deficient (<70) indexes were recorded in 53% of the sample at the WMI, 43% at the PSI, 12% at the VCI, and 6% at the PRI.

### 3.2. Behavioral and Emotional Characteristics

As shown in Figure 1, scores in the Internalizing syndrome scale were significantly higher than in the Externalizing syndrome scales (t(120) = 7.69, *p* < 0.001). Five percent (6/121) of the children presented CBCL DP scores within the clinical range. Partial Eta Squared was 0.33, indicating a large effect size for the general type of symptoms.

Forty-eight percent of the children (58/121) showed borderline or clinical range scores in one or both CBCL syndrome composite scales. Specifically, on the Internalizing syndrome scale, 33% of the children (40/121) had scores in the borderline (7/40) or clinical range (33/40). On the Externalizing syndrome scale, 16% of the children (20/121) had scores in the borderline (12/20) or clinical (8/20) range.

On the Total Problem syndrome scale, 22% (27/121), showed clinical and 12% (15/121) borderline range scores.

A significant difference across the eight syndrome subscales was found (F(6, 672) = 28.32, *p* < 0.001) in the CBCL syndrome subscales. Partial Eta Squared was 0.14, indicating a large effect size. Post-hoc comparisons revealed that Anxious/Depressed, Withdrawn/Depressed, and Social Problems subscales had significantly higher scores than the Thought Problems, Rule Breaking Behavior, and Aggressive Behavior subscales (*p* < 0.05).

The Attention Problem subscale had significantly higher scores than Rule-Breaking Behavior and Aggressive Behavior subscales (*p* < 0.05) (Anxious/Depressed = Withdrawn/Depressed = Social Problems > Thought Problems = Rule Breaking Behavior = Aggressive Behavior, Attention Problem > Rule Breaking Behavior = Aggressive Behavior). In the appendix (Appendix A), t statistics for Bonferroni post-hoc corrections are reported.

Accordingly, as shown in Figure 2, scores in the borderline or clinical range were found in 19% (23/121) of the patients in the Anxious/Depressed subscale, 15% (18/121) in the Withdrawn/Depressed, 15% (18/121) in the Attention Problems subscale, 12% (14/121) in the Somatic Complaints, 11% (13/121) in the Social Problems, 10% (12/121) in the Thought Problems subscales, 4% (5/121) in the Rule Breaking Behaviors, and 3% (4/121) in the Aggressive Behaviors subscales.

A significant difference across the DSM-oriented sub-scales emerged (F(4, 468) = 27.20, *p* < 0.001), with a large effect-size (Partial Eta Squared = 0.18). Post-hoc comparisons revealed that Anxiety Problem subscale showed significantly higher scores than all the other subscales (*p* < 0.05), Affective Problem subscale mean score was significantly higher than the mean scores in Somatic Problems (*p* < 0.05), Oppositional Defiant Problems (*p* < 0.05), and Conduct Problems (*p* < 0.05) subscales. Attention Deficit/Hyperactivity Problem subscale had significantly higher scores than Oppositional Defiant Problems (*p* < 0.05) and Conduct Problems (*p* < 0.05) subscales (Anxiety Problem > all subscales, Affective Problem > Somatic Problem = Oppositional Defiant Problems = Conduct Problems, Attention deficit/Hyperactivity problem > Oppositional Defiant Problems = Conduct Problems). In the appendix (Appendix A), t statistics for Bonferroni post-hoc corrections are reported.

At DSM-oriented sub-scales, scores in the borderline or clinical range were found in 25% (30/121) of patients in Anxiety Problems subscale, 15% (18/121) in the Affective Problem, 11% (13/121) Attention Deficit/Hyperactivity problem subscales, 7% (8/121) Somatic Problem subscale, 3% (4/121) in Conduct Problem Oppositional, and 3% (4/121) in Defiant Problems sub-scales (Figure 3).

### 3.3. Relationships and Interactions among the Background, Learning, Cognitive, and Emotional Characteristics

#### 3.3.1. Relationship with the Background Characteristics

Sex did not affect learning scores, either in terms of the median score of impairment or the number of impairments across measures (z < 1, n.s.). Age significantly correlated with the median impairment score (rho(121) = −0.19, *p* = 0.02) as higher age corresponded to lower impairment. The group with a positive familiarity for neurodevelopmental disorders (26/121, 22%) showed a higher median impairment than that one with a negative familiarity (z = −2.4, *p* < 0.05).

No effects of sex, age, or familiarity for neurodevelopmental disorders were found on the cognitive indexes.

Regarding the CBCL scores, worse scores were found in males than females in the Attentional Problems (F(1, 121) = 7.1, *p* < 0.001) and Rule-Breaking Behaviors (F(1, 121) = 6.08, *p* < 0.01) CBCL subscales. No significant effect of familiarity for neurodevelopmental disorders was found on any CBCL scales and subscales. Higher scores at the Withdraw, Somatic Complaints, and Internalizing scales were associated with higher chronological age (r ranging from 0.18 to 0.26, *p* < 0.05).

#### 3.3.2. Relationship with the Cognitive Profile

Subjects with Mixed SLD showed lower scores in the intelligence indexes, with the exception of PRI, than those with an isolated SLD (IQ: F(1, 113) = 17.3, *p* < 0.001; VCI: F(1, 113) = 10.4, *p* < 0.005; PRI: F(1, 113) = 3.11, ns; WMI: F(1, 113) = 12.9, *p* < 0.001; PSI: F(1, 113) = 8.02, *p* < 0.01). Regression analysis, with Intelligence Indexes (VCI, PRI, WMI, PSI) as predictors and median learning impairment as outcome variable, showed that the intelligence profile of the child significantly predicted the learning impairment (R2 = 0.34, F(4, 113) = 3.7, *p* < 0.001), although only the WMI resulted in being a significant predictor (beta = −0.24, t = −2.41, *p* < 0.005).

Correlational analysis between Intelligence Indexes (i.e., VCI, PRI, WMI, PSI) and CBCL scores showed a significant relationship between WMI and scores in the Rule-Breaking subscale (r = −0.30, *p* < 0.001).

Based on the obtained results, a mediation model analysis was run on the median learning impairment as a dependent variable. As shown in Figure 4, learning impairment is predicted by age and familiarity for neurodevelopmental disorders and, indirectly, via the mediation of the Working Memory Index by the Rule-Breaking Behavior scale.

## 4. Discussion

Specific Learning Disabilities (SLD) are clinical conditions in which difficulties in specific academic areas (reading, writing, and/or mathematics) present high comorbidity with emotional-behavioral disorders. In the literature, the presence of psychopathological difficulties in subjects with SLD varies from about 30% [11] to 70% [60]. The presence of emotional-behavioral problems in SLD could be associated with a series of biological, cognitive, and environmental factors, such as the severity of symptomatology, the late diagnosis, the cognitive level border, the age, and the male sex [3,22,24,27]. However, the number of studies examining these aspects is scarce, thus the type of relationship between these factors and the presence of internalizing and externalizing symptoms in subjects with SLD have to be further investigated.

This study was aimed at investigating the presence and the typology of emotional and behavioral problems in a sample of subjects with SLD and the factors mediating the relationship between learning impairment and psychopathological symptoms. For this purpose, we described the psychopathological profiles at the CBCL [44,45] and their relation with cognitive, learning, and background variables in a sample of 121 children and adolescents with SLD.

The profiles found within the learning, cognitive, and emotional-behavioral areas confirmed and extended the results of previous studies. Regarding the learning profile, it was confirmed that subjects with Mixed SLD or familiarity for neurodevelopmental disorders had higher median impairment than those with isolated SLD disorder or absence of familiarity [17,61]. In addition to previous results, it was found that age was related to learning impairment, as the younger subjects showed worse performances if compared to the older ones. Given that in the present study, the chronological ages correspond to the age of first diagnosis, this result suggests that subjects with milder symptoms require clinical consultation later.

Concerning the WISC-IV/WAIS-IV cognitive profile, Working Memory and Processing Speed Indices were worse than both Verbal Comprehension and Perceptual Reasoning Indices, as shown in the literature [62]. Indeed, impairment of working memory and speed of processing characterized nearly half of the sample, 53% and 43%, respectively. Several studies also documented the influence of working memory and speed of processing on learning disorders and the predictive role of working memory on the degree of learning deficit [63]. In our results, cognitive profiles explain almost 34% of the learning impairment, and the Working Memory Index was a significant predictor.

The analysis of the results of the CBCL showed that about half (48%) of the SLD subjects manifested emotional-behavioral symptoms with a prevalence of internalizing (33%), such as anxiety and depression, over externalizing (16%) disorders. The incidences found are similar to those reported in the literature in SLD [3,60] and higher than in typical development [64,65]. Despite this result, CBCL was not directly related to cognitive profile and learning measures.

The present study enriches the previous literature by investigating the role of cognitive and background characteristics on the emotional-behavioral phenotypes and their relationship with the learning profile in SLD.

Chronological age represents a crucial factor for the display of internalizing problems in SLD subjects: In fact, the manifestation of internalizing symptoms, particularly withdrawal and somatic problems, tends to increase with age. As older subjects showed a milder learning impairment, that result supports further that there is not a direct relationship between learning and internalizing problems in SLD. In adolescence, a learning problem can have a greater impact on mood even in cases of milder disorders and probably the late diagnosis could have exposed the subject to a longer history of school failure, with consequent repercussions on the self-esteem. Sex and cognitive characteristics represent a crucial factor for the expressiveness of externalizing problems: Attentional problems and rule-breaking behaviors are higher in males than females, and emotional problems are related to higher working memory impairment [3,24,28].

One other main contribution of the present study came from the mediation model analysis. Learning impairment was predicted by age and familiarity for neurodevelopmental disorders and, indirectly, via the mediation of the Working Memory Index, by the Rule-Breaking Behavior scale. This result enriches the previous literature as it suggests that behind the acknowledged relation between low working memory skills and learning impairments in SLD, there could be the role of behavioral regulation. Those subjects with greater difficulties in following rules and controlling behavior may pay less attention to maintaining and updating verbal information in working memory with cascade effects on learning competence [66].

Considering the results as a whole, on the one hand, our study confirms the previous literature concerning the greater presence of internalizing problems in subjects with SLD, especially when they are older, and the higher manifestation of externalizing problems in males and in children with low working memory profiles. On the other hand, the present study increases the knowledge about the relationship between externalizing problems and learning impairment, reporting that this relation is not direct but mediated by cross-cognitive processes, such as working memory. Thus, a potential role of working memory as a mediator of the relationship between behavioral regulation and learning impairment may be supposed.

The present study has some limits, such as the inclusion of a widespread age range and the need to collect longitudinal data that could help in describing the developmental trajectories of the emotional-behavioral problems in SLD and their relationship with learning disorders. Another limitation is that the chronological age of our sample overlapped with the age of first diagnosis, thus limiting the interpretation of the age effects. Despite the good psychometrical properties of the CBCL, it is a parental reports diagnostic screening tool, thus future and longitudinal studies are needed to implement other clinical or educational data on the emotional-behavioral characteristic of children and adolescents with SLD.

These results may have implications for clinical practice: Beyond the importance of combining learning assessment with emotional and neuropsychological profile, it is of the utmost importance to provide psychological and emotional support to subjects with SLD, because of the high presence of internalizing problems and the cascade effects of externalizing problems on cognitive and learning profile. Moreover, our results support the growing literature on the implementation of training on working memory [67,68] and on emotional-behavioral problems [69,70,71] in SLD, especially in this clinical condition when behavioral and regulation problems are revealed.

## 5. Conclusions

An emotional-learning-cognitive model is proposed in order to relate the different factors that may characterize the inter-individual variability of individuals with SLD.

Although further studies with larger samples and more extensive neuropsychological investigations may be conducted to confirm these findings, our study is the first attempt to understand how the cognitive profile mediates emotional-behavioral phenotype in SLD subjects.

## Figures and Tables

**Figure 1 jcm-12-01882-f001:**
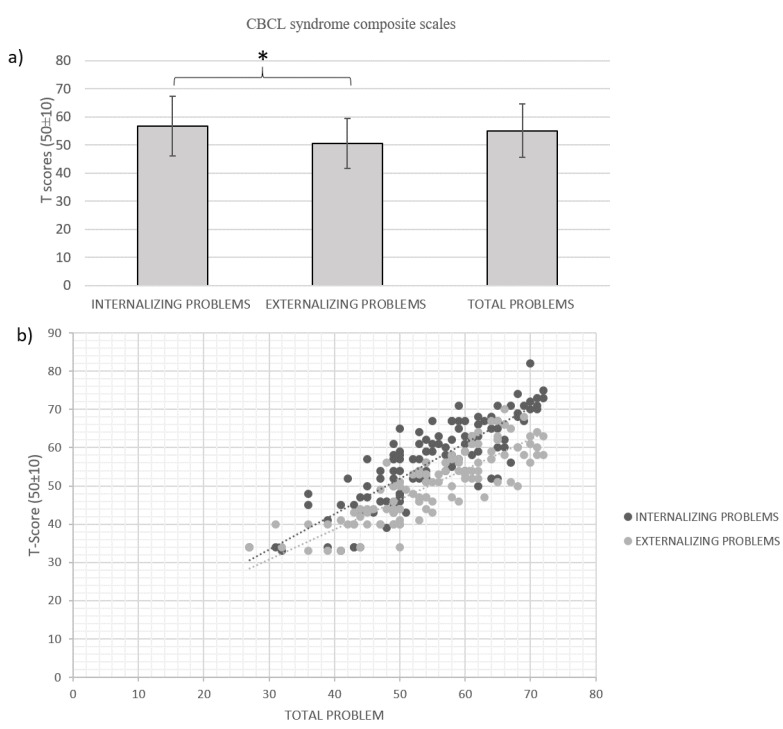
(**a**) Mean and Standard Deviation of T scores in each CBCL syndrome composite scale, * represent the statistical significant differences (p < 0.05) between CBCL syndrome scales; (**b**) scatterplot distribution of Internalizing (black) and Externalizing (gray) Problems scales.

**Figure 2 jcm-12-01882-f002:**
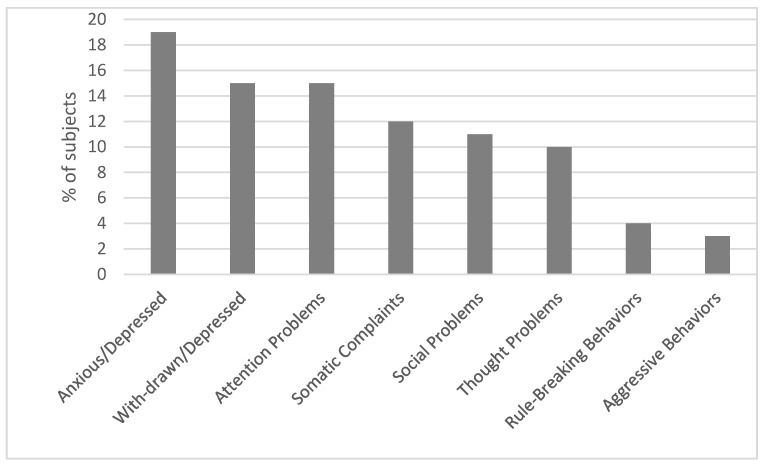
% of scores in the borderline (64–69) or clinical (>70) range at the syndrome subscales.

**Figure 3 jcm-12-01882-f003:**
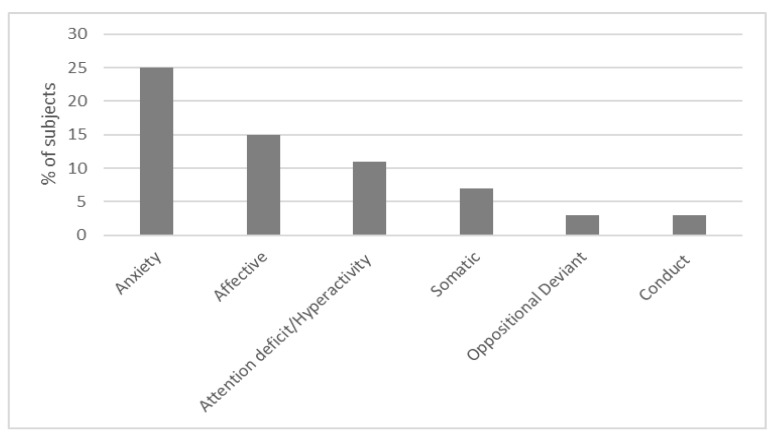
% of scores in the borderline (64–69) or clinical (>70) range at the DSM-5 oriented sub-scales.

**Figure 4 jcm-12-01882-f004:**
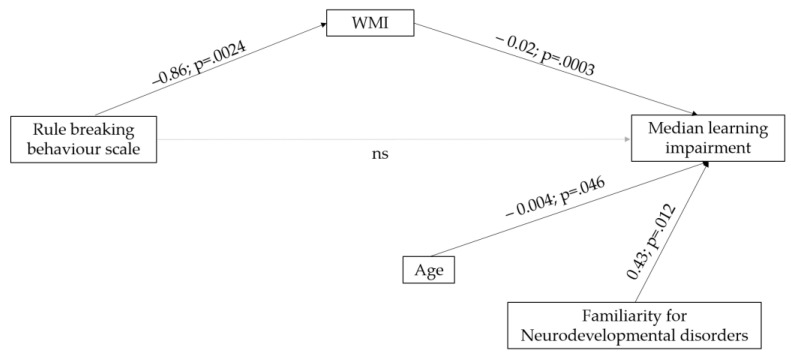
A mediation model analysis on the relationship between background, cognitive, and CBCL variables on median learning impairment (Indirect effect: BootLLCI = 0.0057; BootULCI = 0.03306).

**Table 1 jcm-12-01882-t001:** Descriptive statistics of the score range (1, Normal, 2, Borderline, 3 Deficient) obtained at the academic tests.

	Text Decoding Accuracy	Text Decoding Speed	Text Comprehension Accuracy	Text Dictation Accuracy	Sentences Dictation Accuracy	Math Quotient
Median	2	3	2	3	2	2
Interquartile Differences	2	1	1	1	1	1

**Table 2 jcm-12-01882-t002:** Cognitive profile at the Wechsler scales.

	N	Minimum	Maximum	M	SD
VCI	117	70	124	98.90	11,724
PRI	118	65	141	101.58	13,145
WMI	115	55	115	84.83	13,543
PSI	115	53	138	90.99	13,646

Legend: WISC-IV descriptive statistics of the sample of subjects. VCI: Verbal Comprehension Index; PRI: Perceptual Reasoning Index; WMI: Working Memory Index; PSI: Processing Speed Index.

## Data Availability

The datasets generated for this study are available on request to the corresponding author.

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
