# Peer review of "Specific Learning Disabilities and Emotional-Behavioral Difficulties: Phenotypes and Role of the Cognitive Profile"

_jcm, 2023, doi:10.3390/jcm12051882_

Round 1

Reviewer 1 Report

Thank you for allowing me to review this manuscript. The paper provides additional information about the comorbidity of externalizing and internalizing behaviors among children and adolescents with Specific Learning Disability. Adding important information to the field by also analyzing potential mediating factors involved in the increased prevalence of these behavior problems.  Overall, the paper was well written, however I do have several comments and minor edits for authors to consider. 

Introduction: This was very well-written setting up a strong foundational argument for the current study. Beyond the scope of this paper, but authors may want to investigate other models of SLD identification such as consistency models as opposed to discrepancy models. Paper that gives overview " A Misuse of IQ Scores: Using the Dual Discrepancy/Consistency Model for Identifying Specific Learning Disorders" by Beaujean, Benson, McGill, Dombrowski, 2018.  This may help to inform their work in the future. 

Methods-Methods were well written as well. A few edits here. Line 198 describes the Clinical T-Score range as > or equal to 210. Is that accurate? 

Results-Overall well outlined but a few edits are provided. Line 357 says "but PRI" this sounds potentially imprecise. If I am understanding the sentence correctly, it may bet better to say "with the exception of PRI".  Line 364 need to correct SPI to PSI. Tables 1 and 2-it would be helpful to have a better visual representation of these descriptive tables. I kept wanting to see the range of scores visually to better understand the variability of the sample. If there is another way to convey this information that would be helpful (e.g., scatterplot?). 

Discussion-Well written section-a few typos that require attention. Line 400 says "Speeding Processing" but needs to say "Processing Speed". Line 433 and 438 both say SDL and need to say SLD. The limitation sections could be improved by noting that behaviors were based solely on parent report and not linked to any actual clinical diagnosis nor rounded out with teacher report to speak to level of impairment. 

Conclusion-Line 453 remove "s" from "Characterizes" 

Author Response

Thank you for allowing me to review this manuscript. The paper provides additional information about the comorbidity of externalizing and internalizing behaviors among children and adolescents with Specific Learning Disability. Adding important information to the field by also analyzing potential mediating factors involved in the increased prevalence of these behavior problems.  Overall, the paper was well written, however I do have several comments and minor edits for authors to consider. 

Introduction: This was very well-written setting up a strong foundational argument for the current study. Beyond the scope of this paper, but authors may want to investigate other models of SLD identification such as consistency models as opposed to discrepancy models. Paper that gives overview " A Misuse of IQ Scores: Using the Dual Discrepancy/Consistency Model for Identifying Specific Learning Disorders" by Beaujean, Benson, McGill, Dombrowski, 2018.  This may help to inform their work in the future. 

            Thank you for the suggestion, the current Italian guidelines on SLD recommend using a discrepancy model in the diagnosis of SLD. The paper you suggested to us provides important points for reflection which surely could be developed and analyzed in following studies and research even on our clinical samples.

Methods-Methods were well written as well. A few edits here. Line 198 describes the Clinical T-Score range as > or equal to 210. Is that accurate? 

Thanks for your observations, the Child Behaviour Checklist Dysregulation Profile’s score is not a T-score because it is calculated summing T-scores of some CBCL subscales. Following your suggestion, we modified the sentence.                                                                                  

Results-Overall well outlined but a few edits are provided. Line 357 says "but PRI" this sounds potentially imprecise. If I am understanding the sentence correctly, it may bet better to say "with the exception of PRI".  Line 364 need to correct SPI to PSI. Tables 1 and 2-it would be helpful to have a better visual representation of these descriptive tables. I kept wanting to see the range of scores visually to better understand the variability of the sample. If there is another way to convey this information that would be helpful (e.g., scatterplot?). 

Thanks for your comments, we modified the edits as suggested and we added a scatterplot chart (Figure 1b) to facilitate the visual inspection of the variability of the sample for composite CBCL subscales.

Discussion-Well written section-a few typos that require attention. Line 400 says "Speeding Processing" but needs to say "Processing Speed". Line 433 and 438 both say SDL and need to say SLD. The limitation sections could be improved by noting that behaviors were based solely on parent report and not linked to any actual clinical diagnosis nor rounded out with teacher report to speak to level of impairment. 

Conclusion-Line 453 remove "s" from "Characterizes" 

As suggested, we modified the typos and added in the limitation section the important issue proposed.

Please see the attachment to view manuscript revised

Reviewer 2 Report

Title: Specific Learning Disabilities and emotional-behavioural difficulties: phenotypes and role of the cognitive profile

The current study focuses on an important clinical problem by investigating the emotional behavioral phenotypes of children with SLD, and how age, sex, and cognitive characteristics may influence the relationships between emotional behaviors and learning difficulties. The manuscript is well-written but would benefit from small modifications.

Abstract:

1.       It would be helpful to specify how age, sex, and cognitive functions play as factors/mediators for emotional behaviors and learning impairments. For example, younger children have greater internalizing problems compared to older children, and males have greater externalizing problems compared to females.

Introduction:

1.       The authors did a great job reviewing the literature about the phenotypes of SLD and the potential factors that might influence the emotional-behavioral, and learning difficulties in children with SLD. Could the authors provide hypotheses associated with each research goal based on the literature review?

Methods:

1.       Since the subjects’ age is biased to the age of first diagnosis, I suggest using the age when the subjects received the diagnosis instead of the age at recruitment.

Results:

1.       In Table 2, the number of subjects differed between different Wechsler subscales. Please describe the data missing rates and corresponding reasons in the results sections.

2.       Please provide t statistics for all post-hoc analyses.

Discussion

1.       I suggest moving the clinical implication section from the conclusion to the discussion section and expend it by suggesting specific clinical practices and adding corresponding citations. For example, were there any studies that used working memory interventions to address the learning and emotional behavioral performance in children?

2.       In the last paragraph of the discussion section, the author did a great job listing the limitations of the study. It would be beneficial if the author also provides suggestions for future researchers.

Minor Edits:

1.       Please avoid having less than 3 sentences in a paragraph. Some of the paragraphs could be combined (e.g., Lines 53-61, Lines 95-109, Lines 156-167, Lines 183-192).

Author Response

The current study focuses on an important clinical problem by investigating the emotional behavioral phenotypes of children with SLD, and how age, sex, and cognitive characteristics may influence the relationships between emotional behaviors and learning difficulties. The manuscript is well-written but would benefit from small modifications.

Abstract:

  1. It would be helpful to specify how age, sex, and cognitive functions play as factors/mediators for emotional behaviors and learning impairments. For example, younger children have greater internalizing problems compared to older children, and males have greater externalizing problems compared to females.

Thanks for your comments; we better specified the explanatory factors for emotional and behavioural problems in the abstract. We propose to not insert the explanatory factors for learning problems in the abstract, being not the main aim of the study.

Introduction:

  1. The authors did a great job reviewing the literature about the phenotypes of SLD and the potential factors that might influence the emotional-behavioral, and learning difficulties in children with SLD. Could the authors provide hypotheses associated with each research goal based on the literature review?

Thanks for your suggested, we added in the text some hypotheses associate with each research goal based on the literature review

Methods:

  1. Since the subjects’ age is biased to the age of first diagnosis, I suggest using the age when the subjects received the diagnosis instead of the age at recruitment.

Thanks for your observation, since parents full filled the CBCL during the cognitive and learning assessments, we think the chronological age better represent one of the background data useful for the following analysis. Nevertheless, we are in agreement with your observation and this limitation has been inserted in Limitation section “Another limit is that the chronological age of our sample overlapped with the age of first diagnosis, thus limiting the interpretation of the age effects”

Results:

  1. In Table 2, the number of subjects differed between different Wechsler subscales. Please describe the data missing rates and corresponding reasons in the results sections.

As suggest, we specify in the text the data missing rates and corresponding reasons “Missing data concern the cognitive assessment conducted in another clinical centre. For three subjects all the cognitive scores, for three children the WMI and PSI scales and for one child the CVI scale were not available.”

  1. Please provide t statistics for all post-hoc analyses.

            As suggest, we added t statistics for all post-hoc analyses in Appendix A.

Discussion

  1. I suggest moving the clinical implication section from the conclusion to the discussion section and expend it by suggesting specific clinical practices and adding corresponding citations. For example, were there any studies that used working memory interventions to address the learning and emotional behavioral performance in children?

Thanks for your suggestion, we shifted and expanded clinical implications in Discussion section.  

  1. In the last paragraph of the discussion section, the author did a great job listing the limitations of the study. It would be beneficial if the author also provides suggestions for future researchers.

We added more suggestions for future researches in the discussion section.

Minor Edits:

  1. Please avoid having less than 3 sentences in a paragraph. Some of the paragraphs could be combined (e.g., Lines 53-61, Lines 95-109, Lines 156-167, Lines 183-192).

Thanks for your comments, we combined the sentences in some of the paragraphs as suggested.

Please see the attachment to view the manuscript revised